# Predicting the Non-Deterministic Response of a Micro-Scale Mechanical Model Using Generative Adversarial Networks

**DOI:** 10.3390/ma15030965

**Published:** 2022-01-26

**Authors:** Albert Argilaga, Duanyang Zhuang

**Affiliations:** 1MOE Key Laboratory of Soft Soils and Geoenvironmental Engineering, Zhejiang University, Hangzhou 310058, China; argilaga@zju.edu.cn; 2Center for Hypergravity Experiment and Interdisciplinary Research, Zhejiang University, Hangzhou 310058, China

**Keywords:** micro-scale, constitutive law, asymptotic homogenization, Gaussian Process Regression, Self-Organizing Map, Generative Adversarial Networks, machine learning

## Abstract

Recent improvements in micro-scale material descriptions allow to build increasingly refined multiscale models in geomechanics. This often comes at the expense of computational cost which can eventually become prohibitive. Among other characteristics, the non-determinism of a micro-scale response makes its replacement by a surrogate particularly challenging. Machine Learning (ML) is a promising technique to substitute physics-based models, nevertheless existing ML algorithms for the prediction of material response do not integrate non-determinism in the learning process. Is it possible to use the numerical output of the latest micro-scale descriptions to train a ML algorithm that will then provide a response at a much lower computational cost? A series of ML algorithms with different levels of depth and supervision are trained using a data-driven approach. Gaussian Process Regression (GPR), Self-Organizing Maps (SOM) and Generative Adversarial Networks (GANs) are tested and the latter retained because of its superior results. A modified GANs with lower network depth showed good performance in the generation of failure probability maps, with good reproduction of the non-deterministic micro-scale response. The trained generator can be incorporated into existing multiscale models allowing to, at least partially, bypass the costly micro-scale computations.

## 1. Introduction

Auriault [1] shows that the bulk constitutive response of materials is often determined by their configuration at the micro-scale. In numerical modelling, identifying the cases that need a micro-scale description is important to advance towards more efficient and accurate multiscale models [2]. Some examples of materials with an important micro-structural contribution to the bulk response are listed: masonry structures [3,4], coal reservoirs for methane production [5,6,7], shale rocks [8,9] as well as other rocks [10] and brittle porous materials governing seismic events [11].

In multiscale modelling the constitutive material response is not phenomenologically defined beforehand, instead, at each loading or strain increment the material response is obtained from an underlying numerical model describing the micro-scale e.g., [12,13,14,15,16,17,18]. This approach makes the integration of material points much more costly, with the computational cost often becoming a limiting factor. One of the existing techniques to upscale the micro-scale behaviour to the macro-scale is through asymptotic homogenization [19,20,21,22,23], this technique allows to obtain an equivalent description of the micro-scale in a generic cell and use it as a material constitutive relation e.g., Auriault [24], Argilaga et al. [25]. The constitutive relation can be calculated once at the beginning of the multiscale simulation, or updated at given intervals depending on material evolution allowing to avoid the costly material point integrations. Asymptotic homogenization is only efficient in problems without significant material evolution. In a more general case, real time updates of the material constitutive response are required thus diluting computational economy.

Machine Learning (ML) models are an alternative to the costly micro-scale calculations. Trained ML can computationally advantage classical physics-based numerical methods in several orders of magnitude [26,27,28]. In the geomechanics and materials fields AI (Artificial Intelligence) and ML have shown different levels of success in multiscale problems [29,30], material constitutive modelling [31,32,33,34] as well as in the study of composite in both forward and inverse design approaches [35,36,37]. Some recent applications of AI in the macro modelling of geotechnical problems include: natural hazard prediction and mitigation [38], determination of driven piles bearing capacity in sands using ANN [39], advanced ML techniques [40] and AI systems optimized by evolutionary computation [41], determination of slope stability with ANN [42], among others.

In some cases materials present loss of uniqueness in their mechanical response [43]. Also known as bifurcation, loss of uniqueness is a point in the strain-stress path in which several displacement increments become possible for a unique stress increment. Bifurcation poses several problems in numerical implementations: in first place it degrades performance of zero-finding algorithms e.g., Newton method; those can jump from the neighbourhood of one possible solution to another without ever converging to a solution. In second place, strain localization [44,45] resulting from the lack of a characteristic length in the classical definition of the Cauchy stress [46,47] causes mesh dependency. Numerical methods have been developed to overcome the bad performance of zero-finding algorithms [48] and regularization techniques both in nonlocal [49] and local forms [50,51] address the ill-posedness of the Cauchy expression and mesh dependency. Second gradient model, as a particular case of the Germain theory [51] is one of the available local regularization techniques [52,53,54,55,56,57,58]. Second gradient has been integrated in numerical codes and extensively applied in geomechanics and geotechnics applications with satisfactory results [59,60,61,62]. Despite improvements in zero-finding algorithms and regularization techniques, numerical models both in the micro- and macro-scales can still present bifurcation and instability [63,64].

Because of measurement uncertainties, loss of uniqueness translates into non-determinism in real materials. Numerical models, which are deterministic by default, need to account for non-determinism to properly reproduce natural phenomena. This is usually accomplished by introducing artificial material heterogeneity. Several works study the effects of material heterogeneity and its relation with strain localization [65,66,67]. The introduction of material defects and material heterogeneity in the problem description or numerical discretization are common in geomechanics [68,69,70], these approaches remain very pragmatic and do not question the nature of the material heterogeneity. Indeed, the micro-macro issue of localization onset and its relation with inhomogeneity are still under intense research [71,72,73,74]. General results about possible links between material stability and global stability are still missing [43]. The study of intrinsic material non-determinism as source of heterogeneity is the very subject of the present work. The main novelty is the use of a data-driven ML approach instead of a parametric random field for the reproduction of material heterogeneity.

In this paper, a micro-structure issued from the asymptotic homogenization of a poroelastic matrix with damageable cracks is used to generate a synthetic database of failure states. Normally, this database could be used to build a failure surface delimiting the non-failed space. Unicity and determinism of the solution are evaluated and a series of ML algorithms used to properly reproduce the micro-scale response. The paper is organized as follows: in Section 2 the micro-scale, damage criterion and non-unicity problems are formulated, in Section 3 the different ML algorithms and training results presented, in Section 4 the result from a trained generator is used to provide material constitutive responses. Paper ends with discussion in Section 5 and conclusions in Section 6.

## 2. Problem Description

### 2.1. Micro-Scale Problem

The strong formulation of the micro-scale equations was obtained using asymptotic homogenization [75] in the framework of small strains using multiscale asymptotic expansions [76], full description of the expansions is detailed in Argilaga et al. [25]. The original problem being hydromechanical, the hydraulic and mechanical parts can be uncoupled and only the mechanical part is considered in the present application. A 2D micro-scale structure consisting of a porous elastic matrix and a 1D thin soft layer [77] constituting a crack network is considered. Crack elastic properties are linear in the normal and tangential components and both compression and extension, cracks undergo damage subject to opening. The elementary geometry is periodic in both *x* and *y* dimensions. This micro-scale is representative of materials such as coal, crystalline rocks or composites.

The strong formulation of the problem (mechanical) reads: (1)$${\mathrm{div}}^{y}\sigma^{\left(0\right)}=0\;,\;\mathrm{in}Y$$
(2)$$\sigma^{\left(0\right)}\cdot \vec{n}={\vec{T}}^{\left(0\right)}\;,\;\mathrm{on}\Gamma^{Y}$$
(3)$$\sigma^{\left(0\right)}=c:\left(\epsilon^{x}\left({\vec{u}}^{\left(0\right)}\right)+\epsilon^{y}\left({\vec{u}}^{\left(1\right)}\right)\right)-p^{\left(0\right)}\alpha$$
(4)$${\vec{T}}^{\left(0\right)}=G\cdot \left[\left[{\vec{u}}^{\left(1\right)}\right]\right]-p^{\left(0\right)}\vec{A}\;,\;\mathrm{on}\Gamma^{Y}$$
(5)$${\vec{u}}^{\left(1\right)},\sigma^{\left(0\right)}\;Y-\mathrm{periodic},$$
where *Y* is the elementary periodic cell and $\Gamma^{Y}$ its boundary, $\vec{u}$ the displacement field, in the macro ${\vec{u}}^{\left(0\right)}$ and micro ${\vec{u}}^{\left(1\right)}$ scales, $\sigma^{\left(0\right)}$ is the total Cauchy macro stress tensor, $p^{\left(0\right)}$ the macro pore pressure, *c* the fourth order tensor of elastic stiffness and α the second order tensor of Biot coefficients. Similarly to Equation (Equation 3), Equation (Equation 4) is the Biot equation for the crack network, *G* and $\vec{A}$ are the elastic coefficients of the cracks and its Biot tensor. The stress vector $\vec{T}=\sigma \cdot \vec{n}$ is continuous on the crack jumps and the displacement field $\vec{u}$ discontinuous; discontinuity is the micro displacement field is denoted by $\left[\left[{\vec{u}}^{\left(1\right)}\right]\right]$.

### 2.2. Crack Network Damage

It is considered that damage is concentrated solely in the crack network, this assumption holds true while damage depends on strain and the crack network is significantly softer than the porous matrix. Equation (Equation 4) becomes:(6)$${\vec{T}}^{\left(e\right)}=\left(1-d^{\left(0\right)}\left(\tau \right)\right)G\cdot \left[\left[{\vec{u}}^{\left(1\right)}\left(\tau \right)\right]\right]-p^{\left(0\right)}\left(\tau \right)\vec{A},$$
with:(7)$$d^{\left(0\right)}\left(\tau \right)=\underset{0\leq \rho \leq \tau }{\sup}f\left(\frac{∥\;\left[\left[{\vec{u}}^{\left(1\right)}\left(\rho \right)\right]\right]\;∥}{\Delta_{n}}\right),$$
where *f* is the damage function proposed by Dascalu et al. [78]:(8)$$z\overset{f}{\to }f\left(z\right)=\left\{\begin{array}{cc} z\left(2-z\right) & 0\leq z<1\\ 1 & 1\leq z \end{array}\right.$$
where ${\vec{T}}^{\left(e\right)}$ is the stress in the damaged crack resulting from the asymptotic expansions (e), τ the time-history variable of damage $d^{\left(0\right)}\left(\tau \right)$, ρ the time-history variable of the displacement field ${\vec{u}}^{\left(1\right)}\left(\rho \right)$ in the cracks, $\Delta_{n}$ the length feature of the crack damage.

### 2.3. Numerical Model

The described micro-scale problem is solved using an in-house Finite Element Model. The elementary periodic cell consists of 144 2D 4-node quadrangular elements modelling two types of matrix grains (Figure 1a), node numbering available in the Appendix A, 52 crack nodes modelling the crack jumps (Figure 1b), and 40 linear 2-node elements modelling the crack network (Figure 1c), resulting in a total of 215 nodes. Cracks are initially closed, figure shows cracks with opening for convenience of representation.

Periodic boundary conditions are enforced by penalization, thus increasing the non-zero count in the global stiffness matrix as well as the bandwidth (Figure 2), which in turn increases the computational load of the solver. In order to minimize eigenvector disparity which can cause poor solver performance, penalization stiffness is limited at $10^{3}$ times the stiffness of the undamaged cracks.

Matrix elastic properties are defined by the Lame constants: λ=1.442GPa and μ=0.961GPa (Young’s modulus E=2.5GPa and Poisson’s ratio ν=0.3). These elastic coefficients are borrowed from Marinelli et al. [79] for validation purposes. The problem being non-linear and time-dependent, it is incrementally loaded and secant method is used at each step to find a solution. The loading is macroscopically strain controlled, defined in the space of $\epsilon^{x}\left({\vec{u}}^{\left(0\right)}\right)$, and the response is macroscopic stress in the space of $\sigma^{\left(0\right)}$.

Micro-scale deformation for a loading in each degree of freedom (DoF) is presented (Figure 3), greyscale indicates von Mises stress in Pa. In this case no damage is considered (elastic case), equivalent elastic coefficients of the micro-structure can be obtained via numerical homogenization. The fourth DoF which corresponds to the macro pore pressure has an influence on the micro-structure through the crack network Biot coefficient $\vec{A}$. This influence has proved to be relatively small (see [25], Figure 5), therefore in the following only the first three DoF are considered.

### 2.4. Failure Criterion

A failure criterion is defined accounting for the concept of structural integrity of the overlying mechanical problem [2]. The criterion is as follows: if the force magnitude defined in Equation (Equation 9) decreases a given quantity from a peak, the sample is considered to be failed in the context of the overlying mechanical problem; Equation (Equation 10): (9)$$fm_{c}=∥\sigma_{ij}∥$$
(10)$$fm_{c}<r_{fm}\left(\underset{0\leq \rho \leq \tau }{\sup}\left(fm_{c}\left(\tau \right)\right)\right),$$
where $r_{fm}$ is the peak force ratio. A peak ratio is used because a material point failure does not automatically cause loss of controllability in the overlying mechanical problem [80]. Due to the stiffness of the neighbouring material, the structure can accommodate the new stress distribution without further propagating material damage. A value $r_{fm}=0.8$ is selected as a compromise between high values which fail when reaching the stress peak and low values which create unbreakable structures. Examples of unbreakable micro-structures: https://doi.org/10.6084/m9.figshare.14540118.v1 (posted on 5 May 2021).

The input feature of the model is the loading vector $\epsilon^{x}\left({\vec{u}}^{\left(0\right)}\right)$ and the output feature, which is defined in the same space as the input, is the strain magnitude corresponding to the micro-structural failure. All loadings start at an unstrained configuration i.e., $\epsilon^{x}\left({\vec{u}}^{\left(0\right)}\right)=\left(0,0,0\right)$ and evolve in a linear manner. A sampling of the loading space, either random or systematic allows to obtain the input and output datasets to be used later in the ML algorithm. Assuming determinism of the constitutive response (classical hypothesis), a finite number of failure points can be used to define a boundary between the anelastic region and the failed region in the strain space. This boundary is usually known as failure envelope and parametrized in a variety of failure theories e.g., Mohr-Coulomb, Drucker-Prager or Van Eckelen. A failure envelope of the proposed micro-scale model is generated by applying a succession of equidistant loadings in the space $\epsilon^{x}\left({\vec{u}}^{\left(0\right)}\right)=\left(\epsilon_{xx},\epsilon_{yy},\epsilon_{xy}\right)$, in this example the shear component $\epsilon_{xy}$ is taken equal to zero for sake of visualization. Due to the symmetries of the model the response presents a revolution symmetry which allows to reduce the computational load to one half by only sampling the positive values of $\epsilon_{yy}$. One sample is defined as the input/output pair obtained from a single loading path (Figure 4a). The resolution of the sampling is defined as the number of samples in the loading angle interval [0,π) and in the present example is 64 (Figure 4b).

The 64 sample resolution surface envelope presents a jump in four regions of the *x*–*y* strain space. This results in a non-convex surface which is one of the reasons why meta-heuristics have been used to optimize the problem in previous works [2].

### 2.5. Loss of Uniqueness

Uniqueness can not be assured in problems presenting material softening e.g., [43]. It is proved that when stiffness decreases under loading, the discharge branch can have any slope; and the solution cannot be determined a priori, thus the problem becoming non-deterministic. In a PDE formulation, when any of the eigenvalues of the stiffness tensor becomes zero, the problem loses its ellipticity and any strain applied in the direction of the vanishing eigenvalue fulfils the equilibrium condition, thus making the solution non-unique. Loss of uniqueness, or bifurcation, in numerical models is an interesting phenomenon that agrees with experimental observations; nevertheless, it represents a numerical challenge i.e., decreased efficiency of Newton method, no warranty of finding all possible solutions, main concern in e.g., reliability or failure analysis.

In the following some of the solutions for a biaxial compression test of the presented micro-scale are explored. This is not an exhaustive search to find all solutions but a demonstration of the existence of more than one solution. The asymptotic result of the homogenized stiffness $\sigma_{2222}^{H}$ for a strain $\epsilon_{11}=0.85\%$ is obtained in a parametric study on the number of time steps and tolerance threshold of the Newton algorithm, 2400 computations are performed (Figure 5a). Both vertical and horizontal axes represent discrete values of tolerance and number of steps respectively, thus the plot is not a stress field but rather the representation of individual stress samples. These results showcase the non-uniqueness of the model; infinitesimal variations of the input cause finite variations of the output. Stress values are polarized in two regions, one around $\sigma_{2222}^{H}=2.1\times 10^{8}\;\mathrm{Pa}$ and another around $\sigma_{2222}^{H}=2.6\times 10^{8}\;\mathrm{Pa}$, (Figure 5b). Two loading histories are presented to showcase the first and second solution attractors (Figure 5c).

Parametric test results (Figure 5a) present a pattern of $\sigma_{2222}^{H}$ which seems to follow a random distribution, with the top left part presenting a higher probability of values close to $\sigma_{2222}^{H}=2.6\times 10^{8}\;\mathrm{Pa}$ and the bottom right region close to $\sigma_{2222}^{H}=2.1\times 10^{8}\;\mathrm{Pa}$. In addition, $\sigma_{2222}^{H}$ histogram (Figure 5b) shows that low values seem to concentrate in one narrow region while high values present a distribution in the range $\sigma_{2222}^{H}=\;$[2.60–2.65] $\times 10^{8}\;\mathrm{Pa}$.

Moire Fringe interference plots (Figure 6) discover some patterns in the previous $\sigma_{2222}^{H}=\;$[2.60–2.65] $\times 10^{8}\;\mathrm{Pa}$ stress results in the top left region of (Figure 6a) as well as in a new parametric study with more stringent tolerance values and more samples (Figure 6b), in the latter the pattern appears to depend only on number of steps, not on tolerance threshold. The differences between cases 1 and 2 (Figure 6) indicate that the observed patters are of numerical precision nature and can be predicted by adjusting numerical parameters. Despite the existence of predictable patterns in specific conditions, stress results (Figure 5a) are a demonstration of the non-determinism of the present numerical model. Next section explores micro-scale failure accounting for its non-deterministic nature.

### 2.6. Failure Sampling

A reference micro-scale configuration is used to study the non-unique response according to the criteria defined in Equation (Equation 10). Similarly to the previous Moire Fringe interference study, the number of loading steps and tolerance threshold are used as numerical variability to trigger different solutions. A range between 195 and 205 loading steps (n) is adopted, its value is randomly determined at each simulation with an equal probability for all the range. Tolerance (tol) range is set to follow a uniform distribution between 0.001 and 0.0001. Parameters of the different micro-scale configurations can be found in Table 1 and samples are drawn using a uniform random distribution of the loading angle.

The micro-scale 1 has a standard set of parameters similar to the one used in Argilaga and Papachristos [2], in this case with a stiffer crack network. Micro-scale 2 decreases $\Delta_{n}$ to one half with respect to micro-scale 1. Results of cases 1 and 2, with 400 samples each, present failure points that clearly define a failure surface, a group of points in the vicinity of the surface but scattered in a diffuse region, and some points at a further distance (Figure 7, a and b), the second case (micro-scale 2) presents lower strain values due to the decrease in $\Delta_{n}$. (Figure 7c) shows a detail of the micro-scale 2 in this case with 5000 samples.

Micro-scale 3 (Figure 8) presents a parametric study on physical parameters instead of numerical ones (Table 1, third row). μ coefficient follows a uniform distribution in the range [480–1442] $\times \;10^{6}$ Pa and λ in the range [721–2163] $\times \;10^{6}$ Pa. In this case, the scattering due to the non-unique response of the constitutive law is combined with the physical influence of μ. There is a higher number of samples with lower values of μ further away from the origin, $\epsilon^{x}=\left(0,0,0\right)$, this is due to reaching the state of “unbreakable structure” mentioned before, also lower values of λ show higher strain at failure.

Micro-scales 4 and 5 (Table 1) are not fed with the variation of any parameter, but only with random uniform sampling of the loading angle both in the xx–yy and xx–xy planes. Shear strain $\epsilon_{xy}$ DoF is added in order to obtain solutions in the full loading space (Figure 9). The difference between (Figure 9) a and b is the crack stiffness *G* being 6.5 time larger in the latter, causing many of the samples to represent an “unbreakable structure”.

The last configuration is retained because it generates samples in the three known test fates i.e., failure in the actual failure surface, diffuse samples in the neighbourhood of the surface and points far from the origin due to an “unbreakable structure” phenomenon. A slight modification leads to the creation of the micro-scale 6 (Table 1) by reducing the crack stiffness *G* to $6.0\times 10^{13}$. This is done to decrease the weight of samples with “unbreakable structure” state.

## 3. Failure Modelling

In this section, several ML algorithms are evaluated for the failure modelling of the micro-scale. In order to find the least complex ML algorithm capable of reproducing the micro-scale response, three representative ML candidates with increasing levels of supervision and network depths are selected. Given the characteristics of the micro-scale response, ML candidates are selected to be either good at non-deterministic reproduction, clustering, or both of them.

### 3.1. Gaussian Process Regression

A Gaussian Process Regression (GPR) is a supervised learning method with great flexibility to adapt to unknown data distributions. GPR is a non-parametric kernel-based stochastic model [81], differently from simpler methods such as least-squares, in GPR it is not needed to relate the approximated function to a specific model. This feature makes it suitable for the present application in which the data distribution is a priori unknown. In addition, GPR gives a measure of uncertainty for the predictions which can reproduce the non-determinism of the present application. A GPR model is trained using 1024 samples of the micro-scale 6 in the strain plane $\epsilon_{xy}=0$. The training dataset is kept small because computational time increases cubically with its size due to the required matrix inversion. Predictor data is generated by calculating the loading angle γ, Equation (Equation 11):(11)$$\gamma =atan\left(\frac{\epsilon_{yy}}{\epsilon_{xx}}\right),$$
and the true variable is the norm of the strain vector at failure:(12)$$∥\epsilon ∥={\left(\epsilon_{xx}^{2}+\epsilon_{yy}^{2}\right)}^{1/2}.$$

A hyperparameter optimization is performed using a constant basis function, kernel functions are: Rational Quadratic, Squared Exponential, Matern 5/2, Matern 3/2 and Exponential. The best fit is obtained with the isotropic exponential kernel function (Figure 10).

With fitting metrics: RMSE=0.015452, $R^{2}=0.79$, MSE=0.00023876 and MAE=0.0064823. Despite the relatively good metrics, predictions show many points falling in between the main attractors in real data (Figure 10). In GPR, a Bayesian approach infers a probability distribution over all the data; in the present case, and due to the presence of different failure modes and therefore different distributions, GPR fails at successfully modelling the response. A method capable of identifying the different failure modes is needed in either a stand-alone application or in conjunction with GPR.

### 3.2. Neural Net Clustering

In this subsection neural networks are used to cluster data belonging to the different failure modes. In clustering problems data is grouped according to the similarity of given characteristics. This allows to build a Self-Organizing Map (SOM) which is a low dimensional representation (typically 2D) of a higher dimensional data while conserving its topology [82]. Networks are trained with unsupervised weight and bias learning rules with batch updates i.e., weights and biases are updated at the end of an entire pass through the input data. Network topology consists of one input, one output and one hidden layer with 256 neurons (Figure 11). The input size is 2 corresponding to the strain components $\epsilon_{xx}$ and $\epsilon_{yy}$. A training is performed using 12,000 samples of micro-scale 6.

Net topology in the hidden layer consists in a honeycomb arrangement of 16×16 hexagonal neuron cells (Figure 12a), each cell has six connections with neighbouring cells (Figure 12b), other organizations are also possible. In this case both SOM representation and data are 2D. SOM weight distances map (Figure 12c) presents low values when weights are tightly clustered (yellow cells) and high values separating weights in different feature groups (red and black cells). From this results three categories can be clearly established.

SOM weight positions show the correct identification of the inner surface as a feature cluster (Figure 13a); where green spots are the training vectors, blue spots the neuron’s weights vector and red lines are weight connections or distances. The outmost weights (corresponding to unbreakable micro-structures) are erroneously split between two features. Hits count plot (Figure 13b) shows the feature connections per neuron of the previous. Data evenly distributed across neurons is best for a correct clustering; the highest value of connections per neuron is 286 and neurons between features display zero connections, but overall the data is well distributed within features (Figure 13b).

The scattered group of points in the vicinity of the inner failure surface is not clearly identified as pertaining to a unique feature, therefore the SOM with 16×16 neurons fails at clustering the constitutive response. Other network topologies and training data sizes have been tested with similar results.

### 3.3. Generative Adversarial Networks

Generative Adversarial Networks (GANs) first proposed by Goodfellow [83] is a deep learning approach in which the discriminative model itself is a neural network. Many GANs variants have been proposed since its initial introduction in 2017: Least Squares Generative Adversarial Networks (LSGANs) by Mao et al. [84], Deep Convolutional Generative Adversarial Networks (DCGANs) by Radford et al. [85], Wasserstein Generative Adversarial Networks (WGANs) by Arjovsky et al. [86], among others and they play an important role in the generation of artificial images. A collection of the main GANs codes for Matlab can be found in the repository: https://github.com/zcemycl/Matlab-GAN (accessed on 18 October 2021).

In a GANs a Generator (Gen) consisting in a deep neural network generates samples to imitate the ones of the training distribution while the Discriminator (Dis) gives the probability of the generated samples coming from the generator or from the training dataset. The objective of the training process is for the Gen to maximize the probability of the Dis to make a mistake. At the end of the training the Dis can be dismissed and the Gen fed with random noise $p_{z}\left(z\right)$ to generate synthetic samples. This two-sided minmax game can be written using the value function $v\left(Dis,Gen\right)$ as:(13)$$\min_{Gen}\max_{Dis}v\left(Dis,Gen\right)=E_{x∼p_{data}\left(x\right)}\left[\log\left(Dis\left(x\right)\right)\right]+E_{Z∼p_{z}\left(z\right)}\left[\log\left(1-Dis\left(Gen\left(z\right)\right)\right)\right],$$
where Dis(x) is the probability of *x* coming from the data rather than the Gen distribution $p_{Gen}$. Dis and Gen are simultaneously trained in order to assign the correct tag to data samples and Gen samples (Dis) and to minimize $\log\left(1-Dis\left(Gen\left(z\right)\right)\right)$ (Gen). GANs schematics (Figure 14) depicts two paths of samples going through the discriminator, it is worth noting that the discriminator does not know which one comes from the real data pool and which one from the Gen, only once the prediction is made the nature of the sample is revealed in order to update Dis and/or Gen with the errors and gradients.

Here, the original implementation by Goodfellow [83] is used with some modifications in both Gen and Dis architectures. Both Gen and Dis multilayer perceptrons have been modified by reducing their depth, resulting in four layers of neurons in each network. This reduction is done to avoid the “Helvetica scenario” or mode collapse, a known problem in GANs exacerbated by large numbers of layers and neurons [87]. The architecture of both Gen and Dis is presented in Table 2. Latent dimension is a layer of neurons to accommodate the size of the input data to the Gen architecture, then Gen expands through the layers until reaching the image size. In the other hand, the discriminator starts with an input layer equal to the output of the generator and it narrows down to one single neuron in the output layer (binary real or fake prediction).

Training data consists in 12,000 samples obtained from the micro-scale 6 in the 2D strain space $\left(\epsilon_{xy}=0\right)$. This data is partitioned into 24 segments of 500 samples each and one raster image is obtained from each segment by iteratively converting point coordinates into pixel positions and adding one unit to the corresponding pixel value, finally image values are normalized and can be represented in a linear greyscale. Note that in a GANs the images are converted into a stream of pixel values regardless of rows and columns arrangement, therefore, a string coming from 2D pixels or 3D voxels can be fed to the GANs without any modification of the architecture. In the present work 2D images have been selected for sake of presentation. Hyperparameters resulting from the tuning with the 12,000 sample data are presented (Table 3). GANs code and sample database available in the Appendix A.

The trained Gen can successfully generate images matching the distribution of the 24 ones created with real data. Images are probability maps that, given a strain path, can give the probability of a micro-structure to fail at a given loading level. Assuming a monotonic loading step by step Bayesian conditional probability can also be obtained from the generated probability maps. Real data from micro-scale 6, real probability and generated probability maps are presented (Figure 15).

Real probability sampling pool (Figure 15b) is a probability map of the entire sampling population of 12,000 points or the 24 raster images stacked and normalized in a single one. While the generated probability map (Figure 15c) is one realization of a trained Gen, thus the first does not directly correlate to the latter. GANs results show the ability to reproduce the given sample distribution including the inner failure surface, diffuse failure region and end of loading without failure. The GANs has been properly tuned to avoid mode collapse in the diffuse failure region and several high probability points can be observed (Figure 15c).

## 4. Application to a Material Point Loading

GANs is retained as the best performing out of the three tested algorithms. A GANs generated probability map *I* is used to obtain failure probabilities for given loading paths $\vec{\epsilon }$. A step by step loading is adopted as in the original micro-scale model presented in Section 2. The step size is chosen such as the end of loading is predicted at step 200, this initial estimation is made assuming a homogeneous micro-strain field. The GANs generated image *I* has a noise floor which needs to be filtered to properly identify failure probabilities. A threshold 50% above the RMS is adopted to filter the unwanted image noise. Results for three $\vec{\epsilon }$ slopes are presented (Figure 16), where τ is the history of strain increments.

To refer to the different failure mechanisms, those will be labelled as 1, 2 and 3 from the lower to the higher strain magnitudes. Therefore, failure mechanism 1 is represented by the innermost failure surface, failure mechanism 2 is the diffuse space of scattered failure points, and failure mechanism 3 is not a material failure but the end of loading adopted in the numerical micro-scale model.

Loading case $\epsilon_{22}/\epsilon_{11}=2.0$ (Figure 16, first row) presents a probability density close to $f_{fail}\left(\tau \right)=20\%$ in loading steps 20–23 and $f_{fail}\left(\tau \right)=10\%$ in loading steps 115 and 118. The accumulated probability density $F_{fail}\left(\tau \right)$ is 82% past the failure region 1 and 100% past the failure region 2. Loading case $\epsilon_{22}/\epsilon_{11}=3.7$ (Figure 16, second row) presents a probability density close to $f_{fail}\left(\tau \right)=37\%$ in loading steps 40–41 and $f_{fail}\left(\tau \right)=13\%$ in loading steps 97–98. The accumulated probability density $F_{fail}\left(\tau \right)$ is 73% past the failure region 1 and 100% past the failure region 2. Loading case $\epsilon_{22}/\epsilon_{11}=4.1$ (Figure 16, third row) presents a probability density close to $f_{fail}\left(\tau \right)=24\%$ in loading steps 40–41 and $f_{fail}\left(\tau \right)$ = 15–20% in loading steps 200–202. The accumulated probability density $F_{fail}\left(\tau \right)$ is 48% past the failure region 1 and failure region 2. Probability of reaching end of loading is 52%.

Results highlight the capacity of GANs to provide a failure prediction for a non-deterministic response. Computational economy of the approach can hardly be matched by any numerically-based constitutive model such as FEM or DEM using typical numbers of elements or particles. The avoidance of model collapse, a common issue in GANs, is demonstrated: (Figure 16, first row), where failure probabilities in close-by loading steps 115 and 118 are separated by null failure probability. The approach still suffers from the need to filter the images to eliminate noise; such weakness can possibly be mitigated by the adoption of alternative network architectures and finer hyperparameter tuning.

## 5. Discussion

Results of the different ML algorithms (Table 4) show varying levels of success in reproducing the micro-scale constitutive response. GPR gave good fitting metrics but due to the presence of more than one distribution in the data it was unable to properly reproduce the different failure mechanisms. GPR results can be put down to its own limitations concerning input data. A SOM is able to extract a large number of features from the data given than those features are sufficiently distinguishable. In the present work, SOM failed at properly reproducing some of the scattered regions which are one of the distinctive traits of the micro-scale response. Finally, GANs showed good generation capability for all the failure mechanisms with the advantage of following an unsupervised learning method meaning that no assumptions about input data needed to be made.

The depth of a neural network is a measure of its complexity and the power of the network to perform higher level abstractions from the data. Networks’ depth in GANs, together with their adversarial training approach are two of the attributes that confer them higher cognitive ability. Nevertheless the typical depth used in image generation applications has been reduced in an application of “Ockham’s Razor”; in the XIV century William of Ockham gave the methodological principle (Latin):
*“Pluralitas non est ponenda sine neccesitate”*
or translated into machine learning “models should be no more complex than is sufficient to explain the data”. The principle very much applies to the present work since excessive network complexity in either depth or number of neurons resulted in the “Helvetica scenario” a kind of mode collapse.

Existing surrogates to approximate material constitutive behaviour such as Radial Basis Functions in Surrogate Optimization, among others, cannot reproduce the non-determinism of the approximated function. The proposed approach trains a generator able to provide an unlimited quantity of constitutive response samples as far as it is fed with random noise. The ability to produce infinite non-repeated samples is of great interest in multiscale approaches needing a material description in each of their Gauss points. GANs can also generate material responses in heterogeneous problems by learning the response distribution of each of the constituents. This approach can bridge the gap between phenomenological approaches where heterogeneity is added using a predefined finite set of material properties and multiscale numerical approaches where the size of the problem is limited by computational cost.

The cost of the proposed method is almost entirely caused by the generation of the training dataset; the 12,000 sample dataset used in the SOM and GANs cases required 67 h of CPU time while the training of the networks only a few minutes. The deployment of the trained generator output in a numerical model has negligible computational cost compared to the solver cost in a typical FEM, this is the same situation as with classical phenomenological expressions. Assuming an already available dataset, the economy of the proposed model is assured in almost all cases compared to a multiscale physics-based approach. In the case of requiring a dataset generation, economy of GANs is advantageous as far as the number of required material point evaluations in the physics-based model surpasses the number of dataset samples. Given the typical mesh refinement used in FEM applications and number of loading steps required in nonlinear problems this scenario is easily met and economy raises quickly to a multifold speedup.

## 6. Conclusions

In the present work a series of Machine Learning paradigms have been proposed to provide the surrogate of a micro-scale model response. The main focus being geomaterials, method and results can be generalized to any material. Main conclusions are listed below:Non-determinism is an inherent property of the material mechanical response. It plays a paramount role in the onset end evolution of strain localisation, thus the importance of its reproduction in numerical models.Gaussian Process Regression cannot reproduce the micro-scale response because of the presence of multiple distributions rather than a unique random distribution. This is due to the existence of different mechanisms leading to failure.Neural net clustering using Self-Organizing Maps can distinguish several of the micro-scale failure modes but does not succeed at identifying a small group of data points that are an important feature of the response.With the adequate network architecture, Generative Adversarial Networks can properly reproduce the micro-scale data including its non-deterministic characteristics. As the learning is totally unsupervised, no assumptions about data distribution are needed.The Generator trained in the Generative Adversarial Networks is ready to be deployed in multiscale numerical approaches allowing to, at list partially, avoid the costly numerical micro-computations. The computational economy must allow those multiscale models to compete with classical phenomenological models in terms of cost.

Future research should focus on a full mechanical material description using AGNs i.e., elastic properties, elasto-plastic, visco-plastic and fatigue. Upgrade of AGNs using physics informed networks such as Thermodynamics-based Artificial Neural Networks (TANN) [34]. The $r_{fm}$ coefficient used in the definition of the failure criterion can be enriched with knowledge of the macroscale, this should allow to obtain a holistic multiscale damage model. Data augmentation techniques can help to increase dataset size without requiring additional physics-based simulations. Finally application of the approach in both forward and inverse material design problems and comparison against phenomenological and numerical double-scale models.

## Figures and Tables

**Figure 1 materials-15-00965-f001:**
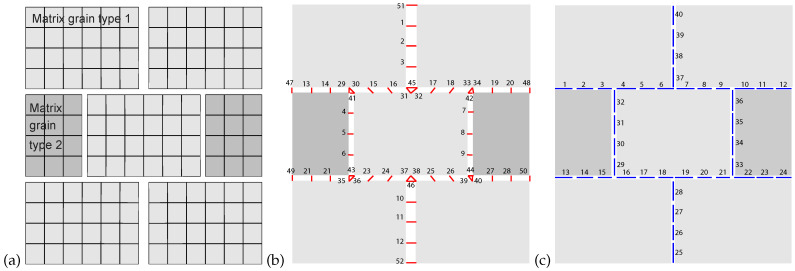
Geometry of the micro-scale Finite Element mesh with two matrix grain types (**a**), crack network nodes or links (**b**) and crack network elements containing two Gauss points each (**c**).

**Figure 2 materials-15-00965-f002:**
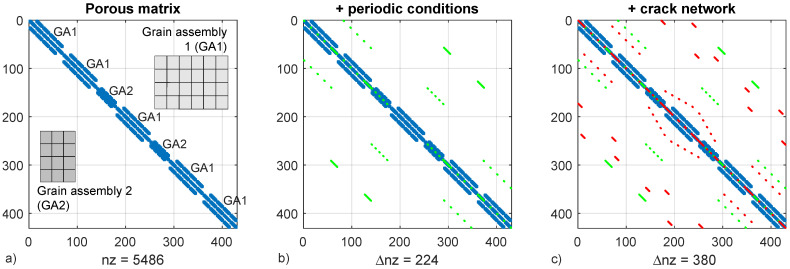
Assembly of the matrix grains in the global stiffness matrix, five grain type 1 assemblies (GA1) and two grain type 2 assemblies (GA2) (**a**). Assembly of periodic boundary conditions and kinematic restrictions by penalization causing a large increase of matrix bandwidth (**b**). Crack network assembly (**c**). Where nz is the non-zero count and Δnz the increase of non-zero count by periodic conditions and crack links.

**Figure 3 materials-15-00965-f003:**
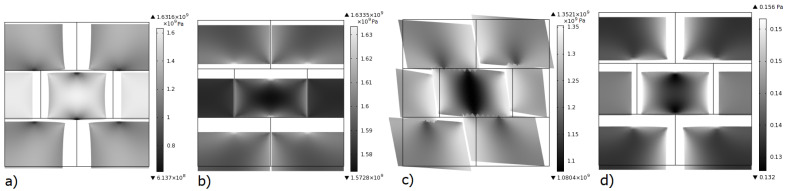
Configurations for macro deformations $\epsilon^{x}=\left[1\;0\;0\right]$ (**a**), $\epsilon^{x}=\left[0\;1\;0\right]$ (**b**), $\epsilon^{x}=\left[0\;0\;1\right]$ (**c**), and $\epsilon^{x}=\left[0\;0\;0\right]$, $p^{\left(0\right)}=1$ (**d**). Colormap represents von Mises stress in Pascals. Crack stiffness: $G=1\times 10^{13}$ Pa. Deformation magnification: 10×.

**Figure 4 materials-15-00965-f004:**
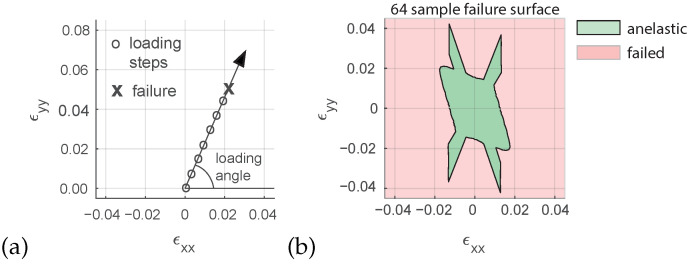
Strain space macroscopic incremental loading (**a**). Failure surface in the xx–yy strain space separating the anelastic non-failed region and failed region, 64 samples resolution (**b**).

**Figure 5 materials-15-00965-f005:**
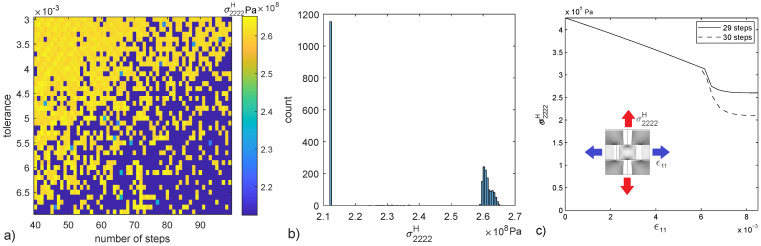
$\sigma_{2222}^{H}$ value for the 2400 cases of the parametric test (**a**), histogram of the 2400 $\sigma_{2222}^{H}$ values (**b**) and stress-strain history for two cases representative of the two main groups of solutions seen in the histogram (**c**).

**Figure 6 materials-15-00965-f006:**
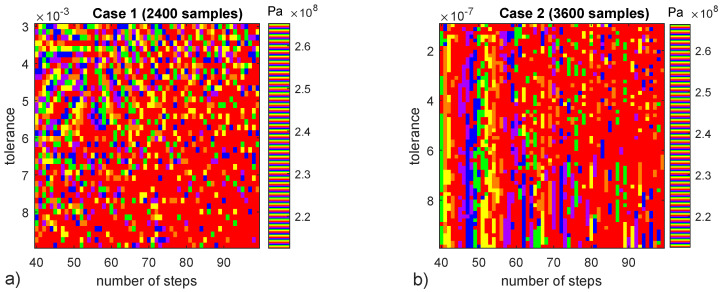
Moire Fringe interference patterns for two cases of number of steps and Newton method tolerance. For some regions of $\sigma_{2222}^{H}$ results seem to follow a predictable pattern while in others there is only random noise. Colormap “prism” in Matalb is used to obtain the interference. (**a**) 2400 sample. (**b**) 3600 sample.

**Figure 7 materials-15-00965-f007:**
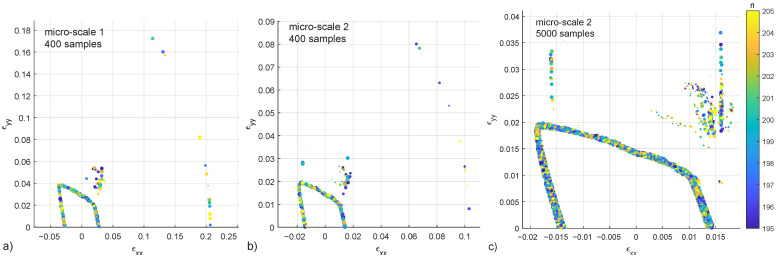
Failure points in the xx–yy strain space for the micro-scale 1, 400 samples (**a**), micro-scale 2, 400 samples (**b**), detail of micro-scale 2, 5000 samples (**c**). Colorbar represents the number of steps [195–205] and the diameter of the dots the tolerance threshold [0.001–0.0001]. Files available in the Appendix A.

**Figure 8 materials-15-00965-f008:**
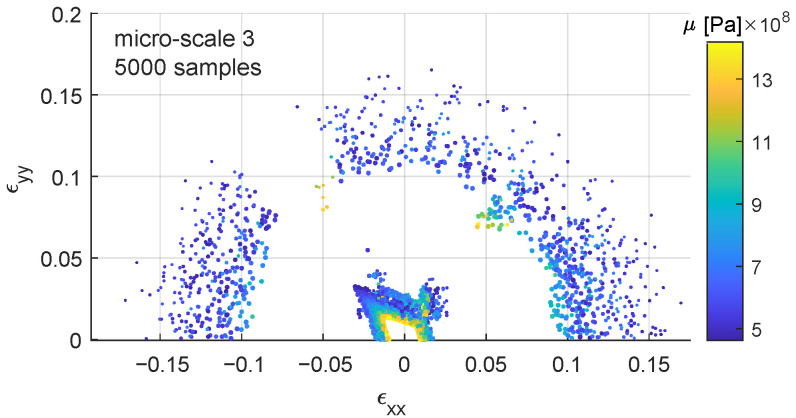
Failure points in the xx–yy strain space. Colorbar represents μ coefficient in the range [480–1442] $\times \;10^{6}$ Pa and the diameter of the dots the tolerance threshold [0.0010–0.0001]. File available in the Appendix A.

**Figure 9 materials-15-00965-f009:**
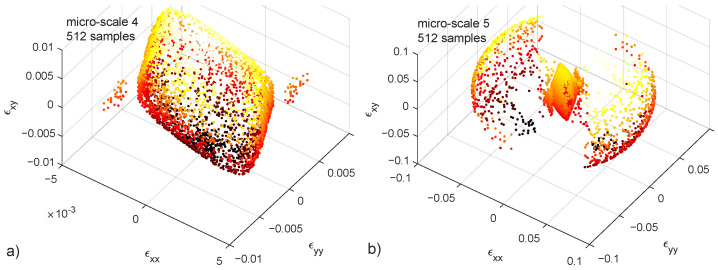
Failure points in the xx–yy–xy strain space. 512 samples in each case. Colorbar represents the value of $\epsilon_{xy}$. Crack stiffness $G=1\times 10^{13}$ (**a**) and $G=6.5\times 10^{13}$ (**b**). 3D figures available in the Appendix A.

**Figure 10 materials-15-00965-f010:**
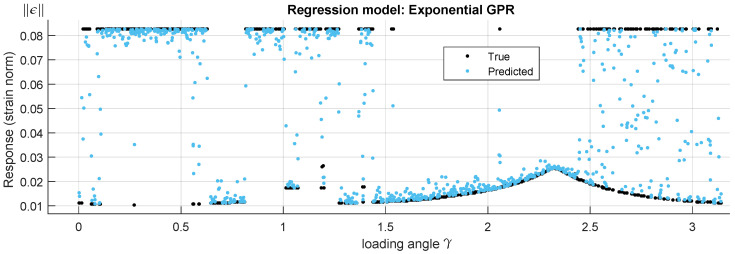
True and predicted response using an Exponential Gaussian Process Regression (GPR) deep learning algorithm.

**Figure 11 materials-15-00965-f011:**
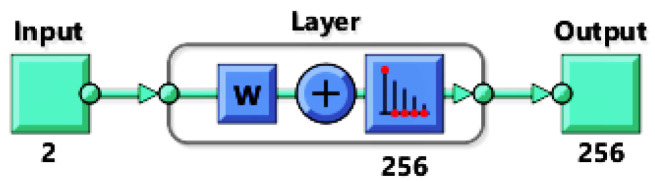
Neural net topology with 16 × 16 neurons in the hidden layer.

**Figure 12 materials-15-00965-f012:**
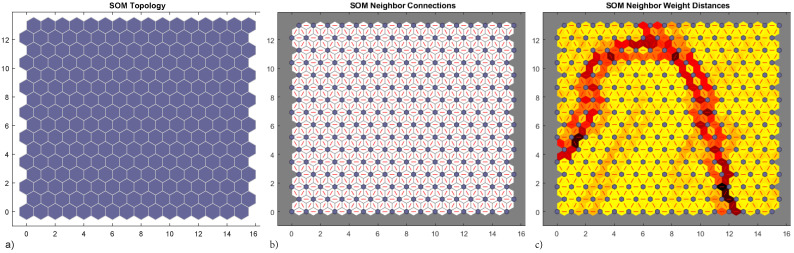
Neural net clustering 12,000 samples, 2D strain $\epsilon_{xx}$–$\epsilon_{yy}$. 16 × 16 topology (**a**), connections (**b**) and weights (**c**).

**Figure 13 materials-15-00965-f013:**
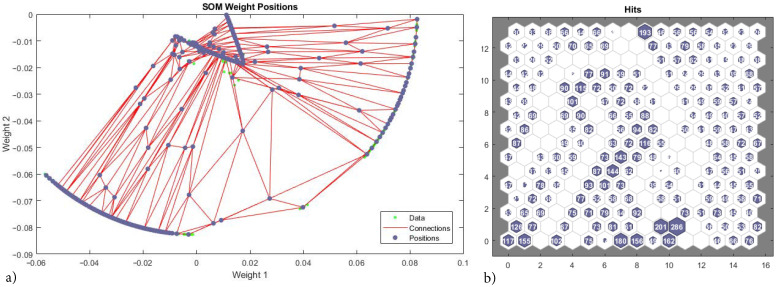
Neural net clustering 12,000 samples, 2D strain $\epsilon_{xx}$–$\epsilon_{yy}$. 16 × 16 topology. Weight positions and hits. (**a**) SOM weight positions. (**b**) Hits count plot.

**Figure 14 materials-15-00965-f014:**
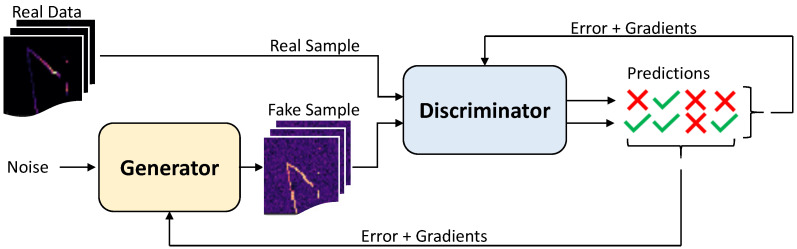
Generative Adversarial Networks (GANs) schematics where the green ticks and red crosses are correct and wrong predictions from the Discriminator respectively.

**Figure 15 materials-15-00965-f015:**
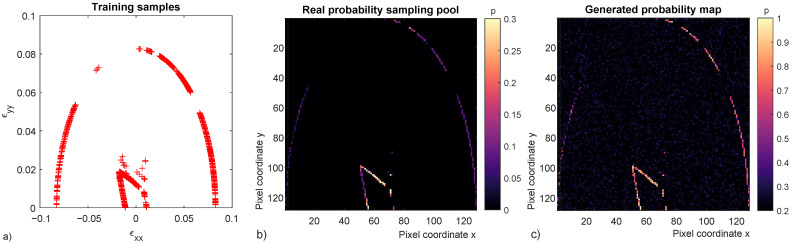
Real data from micro-scale 6 (**a**), real probability sampling pool (**b**) and one generated probability map with GANs (**c**). The three micro-structure fates can be identified in each of the three figures i.e., inner failure surface, diffuse failure region and end of loading without failure.

**Figure 16 materials-15-00965-f016:**
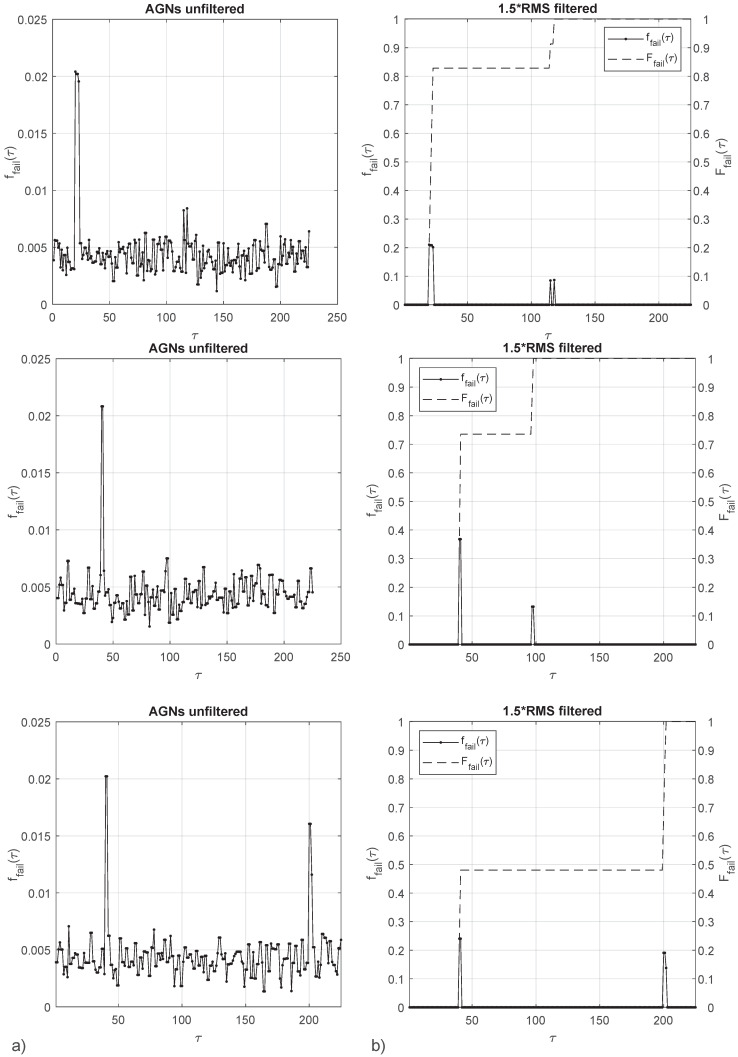
Probability density $f_{fail}\left(\tau \right)$ of a loading profile from a GANs generated probability map (**a**), probability density 1.5×RMS filtered $f_{fail}\left(\tau \right)$ and accumulated probability $F_{fail}\left(\tau \right)$ (**b**). Loading slopes from top to bottom are: $\epsilon_{22}/\epsilon_{11}=2.0$, $\epsilon_{22}/\epsilon_{11}=3.7$ and $\epsilon_{22}/\epsilon_{11}=4.1$.

**Table 1 materials-15-00965-t001:** Micro-scale parameters used in the different parametric studies.

Configuration	$\Delta_{n}$	$\mu \;[10^{6}$ Pa]	$\lambda \;[10^{6}$ Pa]	$G\;[10^{13}$ Pa]	Tol.	n	$f_{\mathrm{mc}}$
μ-scale 1	0.0050	961	1442	$6\times 10^{13}$	[0.001–0.0001]	[195–205]	0.9
μ-scale 2	0.0025	961	1442	$6\times 10^{13}$	[0.001–0.0001]	[195–205]	0.9
μ-scale 3	0.0025	[480–1442]	[721–2163]	$6\times 10^{13}$	0.0001	200	0.9
μ-scale 4	0.0050	961	1442	$1\times 10^{13}$	0.0001	200	0.8
μ-scale 5	0.0050	961	1442	$6.5\times 10^{13}$	0.0001	200	0.8
μ-scale 6	0.0050	961	1442	$6.0\times 10^{13}$	0.0001	200	0.8

**Table 2 materials-15-00965-t002:** Multilayer perceptron architecture of both Generator and Discriminator.

Layer/Neurons	Generator	Discriminator
Input Layer	Latent dimension	Image size
Hidden Layer 1	32	64
Hidden Layer 2	64	32
Output Layer	Image size	1

**Table 3 materials-15-00965-t003:** Hyperparameters tuning.

Hyperparameter	Value
Latent dimension	64
Batch size	24
Image size	128 × 128
Dis learning rate	0.0003
Gen learning rate	0.0003
Gen decay	0.201
Gen squared decay	0.799
Max epochs	150

**Table 4 materials-15-00965-t004:** ML algorithm summary with learning type and neural network depth. A network is said to be deep if it contains two or more hidden layers.

Algorithm	Learning	Depth	Typical Application
GPR	Supervised	Shallow	Curve fitting
SOM	Unsupervised	Shallow	Maps, clustering
GANs	Unsupervised	Deep	Fake image generation

## Data Availability

The data presented in this study are openly available within the document and in the Appendix A.

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
