# Peer review of "Predicting the Non-Deterministic Response of a Micro-Scale Mechanical Model Using Generative Adversarial Networks"

_materials, 2022, doi:10.3390/ma15030965_

Round 1
Reviewer 1 Report
The reviewer entirely agrees that the authors try to solve a quite challenging problem because the non-deterministic response of micro-scale physics is hardly tractable accompanying progressive debonding, buckling, and cracks.
Furthermore, the reviewer also agrees with the research motivation that GAN (Generative Adversarial Networks) can adequately reproduce the micro-scale data, including its non-deterministic characteristics. As the learning is totally unsupervised, no assumptions about data distribution are needed.
Even though it includes good and challenging research content, but its contents are hard to recognize well due to the ambiguity. Thus, you have to explain the questions as follows.
- What is the input and output features, and how to get the data set by biaxial compression test with a numerical model step by step explanation with several figures.
- How can count the sample resolution? It is unclear in Fig. 4, even though they have 64 sample resolutions. That has to be mentioned clearly. The review fails to understand. Also, How can the author get such data in Figure 5 and Figure 6? They have to be mentioned properly.
- In Fig.5, why stress field is plotted like this? The stress field has to be smoothed except the singularity zone.
Finally, the reviewer recommends the authors read previous associated works as follows. By investigating other similar works, authors may have a better idea to predict a non-deterministic response of micro-scale physics as well as how to explain authors data well.
[1] Yongtae Kim+, Youngsoo Kim+, Charles Yang, Kundo Park, Grace X. Gu, and Seunghwa Ryu*, "Deep Learning Framework for Material Design Space Exploration using Active Transfer Learning and Data Augmentation", npj Computational Materials,
[2] S Chakravarty, P Garg, A Kumar, M Agrawal… - arXiv preprint arXiv …, 2021 - arxiv.org
Deep neural networks based predictive-generative framework for designing composite materials
[3] Do-Won Kim, Jae Hyuk Lim, and Seungchul Lee
Prediction and validation of the transverse mechanical behavior considering interfacial debonding of the unidirectional composites through convolutional neural networks, Volume 225, 15 November, 109314, Composite Part B(SCIE), https://doi.org/10.1016/j.compositesb.2021.109314
Reviewer 2 Report
The paper presents a series of Machine Learning paradigms to provide the surrogate of a micro-scale model response. Comprehensive numerical examples are carried out to study the problem. The paper is clear and well-organized. However, the following comments and questions should be addressed prior to recommend for publication.
- The abstract should be revised highlighting the novelties.
- The authors need to improve the quality of figures in the manuscript. The legend of the figures should be also presented clearly and noted with proper symbols, so that the readers can easily understand the meaning.
- How is the computational cost for this proposed method? It should be compared with other methods.
- The language issues can be found locally, improvements must be made.
- Further work on this approach should be mentioned briefly.
Round 2
Reviewer 1 Report
The authors address the issue raised by the reviewer, thus believing that its manuscript has a high qualified contribution to the relevant society.